# Enhancing the Catalytic Activity of Mo(110) Surface via Its Alloying with Submonolayer to Multilayer Boron Films and Oxidation of the Alloy: A Case of (CO + O_2_) to CO_2_ Conversion

**DOI:** 10.3390/nano13040651

**Published:** 2023-02-07

**Authors:** Yong Men, Tamerlan T. Magkoev, Reza Behjatmanesh-Ardakani, Vladislav B. Zaalishvili, Oleg G. Ashkhotov

**Affiliations:** 1School of Chemistry and Chemical Engineering, Shanghai University of Engineering Science, Shanghai 201620, China; 2Laboratory of Adsorption Phenomena, Department of Condensed Matter Physics, North Ossetian State University, Vatutina 44-46, 362025 Vladikavkaz, Russia; 3Geophysical Institute—The Affiliate of Vladikavkaz Scientific Centre of the Russian Academy of Sciences, Markova 93a, 362002 Vladikavkaz, Russia; 4Department of Chemistry, Faculty of Science, Payame Noor University, Tehran P.O. Box 19395-4697, Iran; 5Institute of Informatics, Electronics and Robotics, Kabardino-Balkarian State University, Chernyshevskogo 173, 360004 Nal’chik, Russia

**Keywords:** adsorption, thin films, Mo(110), boron, molybdenum boride, molybdenum boride oxide, adsorption, surface reaction, carbon monoxide, oxygen, surface characterization techniques

## Abstract

In-situ formation of boron thin films on the Mo(110) surface, as well as the formation of the molybdenum boride and its oxide and the trends of carbon monoxide catalytic oxidation on the substrates formed, have been studied in an ultra-high vacuum (UHV) by a set of surface-sensitive characterization techniques: Auger and X-ray photoelectron spectroscopy (AES, XPS), low-energy ion scattering (LEIS), reflection-absorption infrared spectroscopy (RAIRS), temperature-programmed desorption (TPD), electron energy loss spectroscopy (EELS) and work function measurements using the Anderson method. The boron deposited at Mo(110) via electron-beam deposition at a substrate temperature of 300 K grows as a 2D layer, at least in submonolayer coverage. Such a film is bound to the Mo(110) via polarized chemisorption bonds, dramatically changing the charge density at the substrate surface manifested by the Mo(110) surface plasmon damping. Upon annealing of the B-Mo(110) system, the boron diffuses into the Mo(110) bulk following a two-mode regime: (1) quite easy dissolution, starting at a temperature of about 450 K with an activation energy of 0.4 eV; and (2) formation of molybdenum boride at a temperature higher than 700 K with M-B interatomic bonding energy of 3.8 eV. The feature of the formed molybdenum boride is that there is quite notable carbon monoxide oxidation activity on its surface. A further dramatic increase of such an activity is achieved when the molybdenum boride is oxidized. The latter is attributed to more activated states of molecular orbitals of coadsorbed carbon monoxide and oxygen due to their enhanced interaction with both boron and oxygen species for Mo*_x_*B*_y_*O*_z_* ternary compound, compared to only boron for the Mo*_x_*_’_B*_y_*_’_ double alloy.

## 1. Introduction

For a long time, boron-doped materials have attracted much attention in finding wide application in different fields of science and technology, such as material science, mechanical engineering, electronics, optics and others [1,2,3]. Quite recently, a new applied relevance was found as efficient materials for heterogeneous catalysis, often viewed as substitutes to precious metal-containing catalysts [4,5,6,7,8,9,10]. This is due to quite unique transformation of the properties of one or another material upon doping with boron, such as surface atomic reconstruction accompanied by the creation of new active reaction sites, modification of the electron structure favoring adsorption and conversion of molecules and appearance of new reaction channels, etc. For instance, as has been shown very recently by Yang and Wirth [4], boriding of different planes of single crystal tungsten dramatically affects the energetics of both atomic and molecular hydrogen compared to bare tungsten. It is shown that H_2_ adsorption on the W surfaces depends on the surface orientation and the B coverage, and that H_2_ can be chemically adsorbed on the W(100) surface with an adsorption energy of ~−0.5 eV, while the B atoms increase the dissociation energy barrier of hydrogen. Acharya and Turner [5] have shown via density functional theory (DFT) calculations that carbon monoxide (CO) oxidation on Pt(111) supported on boron-doped carbon is notably enhanced compared to pure carbon. This is attributed to the increase of the Pt-support binding energy by approximately 60 kcal/mol. When the effect of the boron-doped carbon decreases upon the growth of the thickness of Pt(111) film, the catalytic properties of Pt with various supports converge after approximately six to nine layers of Pt. Very similarly, Krawczyk and Palczewska [6] investigated catalytic behavior of boron-doped Pd(111). Via a set of in-situ ultra-high vacuum (UHV) surface characterization techniques, they found that selective hydrogenation of hydrocarbons proceeds more efficiently on the boron modified Pd(111) due to Pd(111) surface reconstruction and formation of active reaction sites at the Pd-B interface.

In line with the trend of modification of metals with boron, Tvauri et al. [7] by Surface Science techniques demonstrated that the alloying of Mo(110) substrate with boron dramatically shifts the CO dissociation pathway on bare Mo(110) to oxidation (CO + 1/2O_2_→CO_2_) at the borided Mo(110). It is concluded that this is due to d-band filling of Mo upon alloying with boron via d←p hybridization on one hand, and reconstruction of the surface with exposing the newly formed active adsorption/reaction sites to the gas phase on the other. Apart from being a modifier enhancing the catalytic activity of the corresponding template, the bare boron can also serve as a catalyst, for instance, such as for a nitrogen reduction reaction [8]. By DFT calculations, Liu et al. [8] have shown that hybridized sp^3^ orbitals of boron form bonds by charge backdonation from boron to nitrogen. As a result, the p*-antibonding orbital of N_2_ becomes populated, which causes weakening of the N-N bond. This situation differs from the case of transition metal substrates, when empty d-orbitals, as acceptors of lone-pair electrons of nitrogen, determine the activity of the adsorption/reaction site. Remarkable enhancement of dehydrogenation of hydrocarbons is observed by Ha et al. [9] during combined experimental and theoretical studies of Pt fine particles alloyed with boron. In this case, the decisive role is played by the structures that the Pt_n_B clusters acquire when Pt is alloyed with boron.

Despite quite extensive studies of catalytic behavior of boron-doped materials, the subject of the oxides of corresponding boron-containing materials remains almost totally unexplored. At the same time, the general trend in catalytic activity of materials is that their efficiency increases when shifting from unoxidized materials to the oxides [10]. Taking into account the above consideration, the present work was aimed at determining the effects of oxidation of boron-doped Mo(110) upon the catalytic efficiency of carbon monoxide oxidation on its surface. For this, the experimental Surface Science in-situ UHV studies of model B-Mo(110) and O-B-Mo(110) substrates with well-defined properties for CO oxidation via molecular oxygen have been carried out.

## 2. Experimental

Experimental measurements have been done in an ultra-high vacuum by a set of surface sensitive techniques: X-ray photoelectron (VG Scientific, Birmingham, UK) and Auger electron spectroscopy (XPS, AES) (RBD Instruments, Bend, OR, USA), angle-integrated electron energy loss spectroscopy (EELS) (RBD Instruments, Bend, OR, USA), work function measurements, temperature programmed desorption spectroscopy (TPD) (Hiden Analytical, Warrington, UK), reflection-absorption infrared spectroscopy (RAIRS) (Nicolet, Madison, WI, USA) and low-energy ion scattering (LEIS) (VG Scientific, Birmingham, UK). All these techniques were implemented in the VGS Escalab MkII system (Birmingham, UK) with a base pressure of around 3 × 10^−10^ mbar. For excitation of photoelectrons, a monochromatized Al K_a_ irradiation with photon energy of 1486.6 eV was used, while registering the photoelectrons by conventional 150°-hemispherical electron energy analyzer. The latter was set to a constant pass energy of 15 eV, providing an energy resolution of 0.8 eV, given the used slit width of the analyzer of 6 mm [11]. To detect the LEIS spectra corresponding to scattering of He ions of primary energy of 1 keV, the same analyzer was used. The binding energy scale of the analyzer was calibrated via Au 4f_7/2_ level (84.0 eV) detected from ca. 4 nm thick Au film in-situ deposited onto Mo(110) substrate by thermal evaporation. For very precise determination of the value of the binding energy, a number of parameters, such as the sample work function and its possible variation during measurements, and the alignment of energy levels between the analyzer and the sample, are to be taken into account [12,13]. In the present case, however, the main focus was not the precise detection of the binding energy, but its relative variation upon reactions under investigation, so there was no need to take into account the mentioned parameters. For TPD, the quadrupole mass-spectrometer (Hiden Analytical, Warrington, UK) with the linear temperature sweep of the sample at a growth rate of about 2 K/s was used. During the measurement, the sample was placed at a distance of about 5 mm against the entrance slit of the spectrometer. The spectrometer was tuned for simultaneous detection of CO and CO_2_ (*m*/*z* = 28 and 44 a.u.m.). No background processing procedure of the TPD spectra was applied. The work function was measured by the Anderson method, utilizing a low-energy electron beam via detection of shift of the retarding curves, determined by the contact potential difference between the cathode of the corresponding low-energy electron gun and the sample. The value of the work function of Mo(110) was assumed to be 5.0 eV, taken as a reference point, since the Anderson method allows detection of only the relative change of the work function, not the absolute value. Fourier-transform infrared (FTIR) spectra in RAIRS mode were registered with the aid of conventional IR-spectrometer (Nicolet-870). To detect the vibrational modes of the molecules adsorbed on surface of the sample, the IR-source and the detector (MCT) were aligned across the sample in a way to ensure grazing incidence of the beam (incidence angle is 80 degrees). For the p-polarized light, using such a geometry provides maximal sensitivity to the molecular vibrations aligned along the surface normal. Such configuration is used because the diatomic molecules, like CO, NO, etc, usually adsorb in an upright, sometimes tilted geometry [14]. Auger spectra were detected in dN/dE mode using a single-pass cylindrical mirror analyzer with primary electrons generated by coaxial gun, operating at an energy of 3 keV.

To prepare the B-Mo(110), the boron film was initially deposited onto Mo(110) by electron bombardment. The system was then annealed at a temperature of 1200 K, resulting in a quite stable compound with an amorphous surface structure. The amount of deposited boron atoms on Mo(110) has been in-situ determined by quartz microbalance installed coaxially to the Mo(110) at a sample holder in UHV. The coverage of boron atoms defined as 1 monolayer (1 ML) corresponds to surface concentration of atoms of 2.75 × 10^15^ cm^−2^. For preparation of the B-Mo(110) compound and its oxide, the initial coverage of deposited boron was 2 ML. This film is quite unstable, and its annealing leads to dramatic diffusion of the boron from the surface into the bulk of the Mo(110) substrate. The amount of boron atoms remaining at the surface and subsurface region of few atomic layers is such that the mean B to Mo atomic concentration ratio is 1 to 3.

To realize the adsorption of gaseous species on such prepared samples, the UHV chamber was controllably backfilled by the carbon monoxide and molecular oxygen of research-grade quality via the high-sensitive leak valves. As a measure of the amount of adsorbed molecules, the value of exposure defined as multiplication of the partial pressure and the time during which this partial pressure is kept is used. An exposure unit (1 L) was defined as: 1 L = 10^−6^ mbar × 1 s. The sample was mounted on the manipulator, enabling movement in UHV in (*x*,*y*,*z*)-directions, tilting and 360°-rotation. The sample could be cooled down to about 90 K via a liquid nitrogen reservoir, and heated up to 2700 K by electron bombardment with temperature control by the calibrated W-Re thermocouple.

## 3. Results and Discussion

Before alloying of B with the Mo(110), it is informative to verify the growth mode of the B submonolayer to multilayer film and the electronic state of adsorbed boron atoms on Mo(110) surface held at room temperature. The growth mode of B on Mo(110) in the submonolayer region was determined by the well-known procedure of plotting the Auger uptake curves of the substrate (Mo(110)) and the deposited layer (B) [15]. The corresponding plot for the Mo MNV Auger transition at 188 eV is shown in Figure 1a (curve 2). Its linear decreases with increasing the coverage of deposited boron atoms points at the layer-by-layer growth mode, when the two-dimensional uniform monolayer of boron atoms is formed. From the energetic viewpoint, this means that the energy of interaction between B and Mo atoms is higher than the lateral B-B interaction. Otherwise, this would result in the formation of 3D-islands upon boron growth on Mo(110) in the submonolayer region. Such a B-Mo interaction occurs via the valence charge polarization of Mo towards B atoms. This is concluded from the results of work function measurements presented in Figure 1a (curve 1), where the work function versus coverage plot is shown for submonolayer growth of boron film on Mo(110). The value of the work function of B-Mo(110) gradually increases alongside increasing the B coverage, and stabilizes at a coverage close to 1 ML at a value of 5.8 eV. The growth of the work function indicates that there is charge redistribution at the B-M(110) interface resulting in the net flow of the electron charge from the Mo to B, thus leading to the increase of the work function. Taking into account the well-known approach of using the Helmholtz equation (Δφ = 4πnμe, where n is surface concentration of adatoms, μ —dipole moment) to estimate the dipolar moment of such a charge redistribution at the interface, the dipole moment of a single B atom adsorbed on Mo(110) at n→0 is 0.2 D. This rather small value of m gives an evidence that the chemisorption of B on Mo(110) is rather of the covalent than the ionic nature. This is corroborated by the EELS data obtained during submonolayer growth of boron film, summarized in Figure 1b. It is seen that the surface plasmon intensity (10.5 eV) dramatically decreases as boron coverage grows, and disappears at 0.8 ML, whereas the bulk plasmon loss (24 eV) remains almost unchanged (Figure 1b). When boron coverage exceeds 1 ML, the spectrum consists of one broadened line attributed to the bulk plasmon loss of boron (26 eV). The similarity of the electronic structure of 2D-layer and bulk boron was quite recently also observed by Preobrajenski et al. [16] for boron films on Al(111) by a combination of photoelectron, STM and DFT studies. It is shown that the feature of boron films separates electronic states into s- and p-subsystems, determining the 2D-nature of the electronic system of boron. Transformation of the spectra in the bulk plasmon region can be interpreted as superposition of decreasing of the Mo bulk plasmon and emerging of the boron bulk plasmon loss. Assuming the three-fold hollow of Mo(110) plane as an adsorption site of B, in line with the case of the oxygen-Mo(110) system [17], and taking into account that the boron atom is as twice as small than the Mo atom (2.3 Å versus 4.2 Å), it is reasonable to expect that the boron film is almost coplanar with the topmost Mo(110) layer. This violates the lateral integrity of the Mo electron subsystem responsible for surface plasmon [18]. A rather small initial dipole moment of B (0.2 D) is additional evidence of a negligible shift of boron outwards from the Mo(110) plane as a result of directed covalent chemisorption bond. The formation of a directed bond for boron adsorbed on W(110) is also predicted by Dorfman et al. [19] by atomistic simulations of the deposition process with non-empirical potentials. As an alternative view, it has quite recently been discovered the effect of a band gap opening in two-dimensional monolayer boron films for both the in-plane s + p*_x_*_,*y*_ and p*_z_* orbitals, leading to damping of the surface plasmon intensity [20].

The tendency of quite strong interaction between B and Mo(110), seen by the above transformation of the surface plasmon mode, is further manifested when the system is annealed. As seen in Figure 2, featuring the Auger intensity and the work function changes with the annealing temperature, even quite slight heating (450 K) results in dramatic changes as a result of boron diffusion into the Mo(110) bulk (part ab). Further thermal treatment in the temperature range (700 K–1500 K) causes moderate changes in terms of boron interdiffusion (part bc), followed by more notable changes with the temperature growth up to 2000 K (part cd). As seen from the changes of the Auger spectra with the annealing temperature, shown in an inlay of Figure 2, there is no change of the energy of Auger lines of both boron and molybdenum. However, the position of Mo 3d-photoelectron line shifts by 0.6 eV to lower binding energy after alloying with boron (Figure 3). This is quite direct evidence of a chemical bond formation between boron and Mo upon annealing. The change of electron charge at atomic levels of Mo and B leads to the change of their binding energies, manifested by the observed XPS line shift. The fact that such a shift is not seen in the Auger spectra is presumably due the opposite compensation of chemical and final state effects in the case of an atom with two electron vacancies created in the Auger process. Quite strong interaction in the B-Mo system was also recently demonstrated by dynamical LEED and DFT calculations by Hossain et al. [21] for a boron layer annealed on Mo(112) surface. Furthermore, very similar behavior of boron films in terms of diffusion into the Si(111) substrate bulk accompanied by chemical bond formation is reported by Krugener et al. [22], using ultraviolet photoelectron spectroscopy and reflection high-energy electron diffraction.

Given the different rates of boron interdiffusion in the Mo(110) (Figure 2, parts ab and bc), it is informative to study the diffusion kinetics to get insight into the mechanism of formation of B-Mo(110) alloy. For this, the Auger intensity of B KLL for four similarly prepared B-Mo(110) systems was plotted versus the annealing time, while the B-Mo(110) system was kept at a certain temperature. The corresponding plots are shown in Figure 4. Redrawing these dependences in a semilogarithmic plot of the time (t), after which the B Auger intensity dropped to the same certain level (by 65%), versus the reciprocal temperature, gives the activation energy of boron diffusion into the Mo bulk (inlay). There are two distinct values, which are 0.4 eV for the annealing temperature interval 450–600 K (part ab), and 3.8 eV for 800–1400 K (part bc). The former is likely due to the diffusion of boron atoms from the Mo(110) surface into its bulk without notable interaction between B and Mo atoms, while the latter to the chemical interaction with formation of molybdenum boride. The higher value of activation energy (3.8 eV) of boron concentration decay is in this case attributed to the B-Mo bond breaking and subsequent diffusion of B into the Mo(110) bulk until the boron totally disappears from the surface region at a high temperature (part cd).

In order to probe the outermost surface layer of B-Mo(110) system, the LEIS spectroscopy, which is well suited for this purpose, was used. For more unambiguous interpretation of the spectra, initially the spectra of bare Mo(110) and the thick boron film were recorded. The corresponding spectra are shown in Figure 5 (spectra 1 and 2) with the He^+^ ion scattering features at 807 eV for Mo and 348 eV for B. The latter corresponds to the situation when the entire Mo(110) surface area is covered by the B. After annealing of such a boron film at a coverage of 2 ML at a temperature of 1200 K for 3 min, the observed LEIS spectrum is shown in Figure 5 (spectrum 3). Appearance of the spectral feature at 807 eV, corresponding to Mo, indicates that the part of boron atoms, initially located at the Mo(110) surface, diffuses into the bulk upon annealing, thus exposing the part of the substrate atoms. This is accompanied by the initial decrease of boron LEIS intensity, as seen by the comparison of the corresponding spectral features at 348 eV of spectra 2 and 3 (Figure 5). A comparison of the boron LEIS intensities before (spectrum 2) and after (spectrum 3) annealing indicates that the fraction of B atoms remaining at the outermost layer of the system after annealing is about 1/3 of the monolayer concentration. As has been noted above, the obtained system corresponds to quite stable molybdenum boride [7].

In terms of catalytic performance, for instance, for carbon monoxide oxidation by molecular oxygen, this system is much more active, compared to bare Mo [7]. A further increase of catalytic activity of molybdenum boride is achieved when it is oxidized. The oxidation of the B-Mo(110) system, corresponding to spectrum 3 (Figure 5), was carried out via backfilling the UHV chamber by molecular oxygen up to partial pressure 10^−6^ mbar while keeping the substrate temperature at 900 K for 10 min. The corresponding LEIS spectrum is shown in Figure 5 (spectrum 4). There are two features, differentiating the spectra 4 and 3: (1) a new LEIS band at an energy of 476 eV appears, which is likely attributed to oxygen; (2) Contrary to expectations that the LEIS intensity of boron must drop due to the shielding of boron atoms by the subsequently adsorbed oxygen, the B LEIS feature increases in intensity. For clarity, direct comparison of the corresponding LEIS intensities, demonstrating this trend, is shown in an inlay of Figure 5. Such an increase of the B LEIS intensity upon molybdenum boride oxidation can be explained, assuming that the part of boron atoms, located at the subsurface region, diffuses out to the outermost surface layer, driven by the interaction of oxygen with the molybdenum boride and its oxidation. In favor of this assumption is the above observed B 1s and Mo 3d_5/2_ photoelectron lines shift upon oxidation from 190 to 191 eV, and from 228 to 229.5 eV, respectively (Figure 3). According the work of Schmitt et al. [23], the most favorable ternary Mo-B-O compound formed in this way is molybdenum borate Mo_2_B_4_O_9_. It is this compound which is characterized by a notable relative concentration of boron, and featuring the enhanced diffusion of boron species from the bulk of molybdenum boride to its surface during oxidation. At the same time, taking into account the reduced dimensionality of the surface region, it is quite reasonable to assume the formation of a non-stoichiometric surface Mo_x_B_y_O_z_ compound rather than the stoichiometric molybdenum borate Mo_2_B_4_O_9_.

The adsorption properties of the molybdenum boride and its oxide (borate) are dramatically different. This is seen from the comparison of vibrational spectra of adsorbed carbon monoxide molecules (Figure 6, spectra 1, 2). There is a dramatical red-shift of CO intramolecular vibrational stretch frequency by 151 cm^−1^ on the oxidized alloy; additionally, its intensity notably drops. At least two reasons may be responsible for such spectral features: Wavenumber red-shift and an IR intensity decrease are due (1) to the more tilted adsorption geometry of CO molecule on the boride oxide compared to an unoxidized substrate; and (2) to the enhanced backdonation of electron charge from the substrate to the anti-bonding 2p* molecular orbital of CO (in accordance with the Blyholder model of molecular chemisorption [24]). According to the work of Schmitt et al. [23], the six bonding molecular orbitals in tetrahedral [Mo]_4_ cell are filled by twelve delocalized electrons, which can be easily trapped by the energetically close 2p* antibonding CO molecular orbitals. This leads to the weaking of CO bonding, and thus to the observed red-shift of the CO intramolecular stretch frequency. The IR band changes further when the CO is adsorbed on the Mo_x_B_y_O_z_ surface pre-dosed by oxygen to saturation exposure of 50 L; the intensity decreases even more compared to un-dosed borate (spectrum 2), while the wavenumber shifts to blue by 20 cm^−1^ (spectrum 3). The observed drop of IR intensity is due to the situation when the CO molecules acquire more tilted adsorption geometry on the oxygen pre-dosed surface, compared to bare Mo_x_B_y_O_z_. The indicated blue shift of CO frequency results from the partial transfer of the electronic charge from the 2p* antibonding orbital of CO to more electronegative adsorbed oxygen. This in turn leads to somewhat strengthening of the intramolecular CO bond, and thus to stretch frequency blue-shift. In this regard, it is noteworthy that the behavior of adsorbed CO on the molybdenum boride oxide surface is considerably different from that occurring on the surface of molybdenum boride with the adsorbed oxygen, as was reported earlier [7]. In the latter case, the adsorbed CO frequency is 2067 cm^−1^, which is 142 cm^−1^ higher than the value corresponding to the Mo_x_B_y_O_z_ surface (Figure 6, spectrum 2). Such a dramatic difference points at different adsorption and interaction mechanisms of CO with oxygen at these two types of substrate.

This difference is also manifested in the case of temperature variation for these substrates, revealed by TPD measurements. The corresponding spectra for (CO + O_2_), resulting in CO_2_ formation and its desorption into the gas phase, on the surface of unoxidized molybdenum boride (spectrum 1′) and the molybdenum boride oxide Mo*_x_*B*_y_*O*_z_* (spectrum 2′), are shown in an inlay of Figure 6. There is a distinct difference that in the latter, the CO_2_ formation efficiency is as twice as high than in the former case. Moreover, in the latter case, the desorption temperature of the formed CO_2_ by the (CO + O_2_) reaction via Langmuir-Hinshelwood mechanism on the surface of ternary oxide is 21 degrees lower, also pointing at its higher catalytic activity. According to DFT studies of Stampfl and Scheffler [25], such an increase of activity may be due to the fact that the molecular axis of CO is more tilted to the surface plane, and thus is closer to the coadsorbed oxygen, leading to a lowering of the oxidation reaction barrier. This is evidenced by the fact that the CO vibrational intensity is notably lower for the boride oxide than for bare Mo boride (Figure 6, spectra 1 and 2, respectively). Despite this, the TPD CO_2_ intensity is higher for the molybdenum boride oxide (inlay). Another reason for the inverse proportionality between the intensities of the vibrational lines of CO and the TPD spectra can be the fact that, unlike the Mo*_x_*B*_y_*O*_z_*, a smaller part of CO molecules is converted into CO_2_ on the surface of molybdenum boride. Moreover, the efficiency of oxidation of CO by coadsorbed oxygen is higher for the lower energy of chemisorption of oxygen with the substrate [25]. Comparing the molybdenum boride and its oxide from this standpoint, one can assume that the chemosorbed oxygen is more weakly bound to the molybdenum boride oxide, since the orbitals of Mo and B are already bound to the oxygen of the oxide. This is evidenced by the above-mentioned shift of binding energies of photoelectron lines of Mo and B upon molybdenum boride oxidation. Thus, alloying of Mo with boron, followed by the oxidation of the molybdenum boride formed, can be viewed as a route of increasing the catalytic activity of the substrate for carbon monoxide oxidation. This is due to the change of the structure and morphology of the substrate surface upon alloying and oxidation, leading to the appearance of new adsorption/reaction sites on one hand, and to the overall change of the electronic structure of the substrate, affecting molecular orbitals and their interaction on the other, according to the situation of synergistic effects of metal alloying overviewed very recently by Wang et al. [26]. The catalysts prepared in this way with quite high activity, at least for CO oxidation, may serve as a point of the fundamental background of design of alternatives to noble metal-containing catalysts, which also exhibit similar trends [27,28,29]. The advantage of such catalysts is their low cost and the ability to tune their activity and selectivity by easy changing of their chemical content, structure, morphology and the local and total electron structure.

## 4. Conclusions

Keeping in mind the tendency of tuning the catalytic efficiency of materials by their doping with different atoms, the Mo(110) was modified by boron as one of the recently discovered promising additives. To form such a mixed B-Mo(110) compound, first the boron submonolayer to multilayer films, in-situ deposited onto Mo(110) substrate and held at room temperature, were studied by a set of surface characterization techniques in an ultra-high vacuum. The chemisorption bond formed at the B-Mo(110) interface dramatically changes the electron state of the outermost layer of the substrate, leading to the total dampening of its surface plasmon excitation. The latter, along with the quite small dipole moment of single adsorbed boron atoms (0.2 D), points at the formation of preferentially covalent chemisorption bonds. This can be viewed as a precursor state for B intermixing in the Mo bulk, and the formation of a borided molybdenum upon the annealing of the boron overlayer. Such a double compound is completely different from the bare Mo(110), in that in the latter case, the only reaction of CO on the surface is dissociation, while in the former the dissociation shifts to an oxidation reaction pathway with quite notable activity. The CO oxidation activity considerably grows further when the borided molybdenum is oxidized.

## Figures and Tables

**Figure 1 nanomaterials-13-00651-f001:**
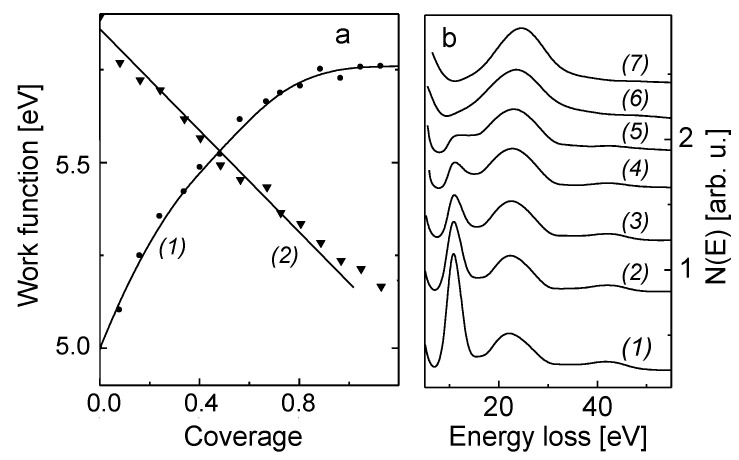
(**a**) Dependence of the work function (curve 1) and intensity of Mo MNV Auger transition, 188 eV (curve 2) upon the coverage during boron submonolayer film growth on Mo(110) held at room temperature. (**b**) Dependence of the EELS spectra upon the boron coverage (ML): Electron energy loss spectra transformation with boron coverage increase (ML): 1–0; 2–0.1; 3–0.2; 4–0.3; 5–0.4; 6–0.8; 7–1.2.

**Figure 2 nanomaterials-13-00651-f002:**
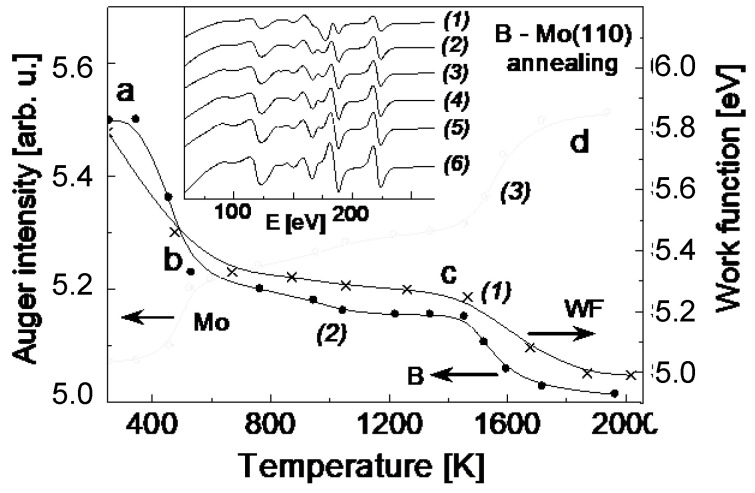
The work function (1) and Auger intensity of B (curve 2) and Mo (curve 3) versus annealing temperature plots of boron film on Mo(110) surface at initial boron coverage of 2 monolayers. Inlay: The Auger spectra upon annealing for 15 s at each temperature indicated. Annealing temperature (K): 1–300; 2–500; 3–600; 4–1150; 5–1500; 6–1750. The spectra are measured at about room sample temperature. Sections (ab), (bc) and (cd) indicate annealing intervals.

**Figure 3 nanomaterials-13-00651-f003:**
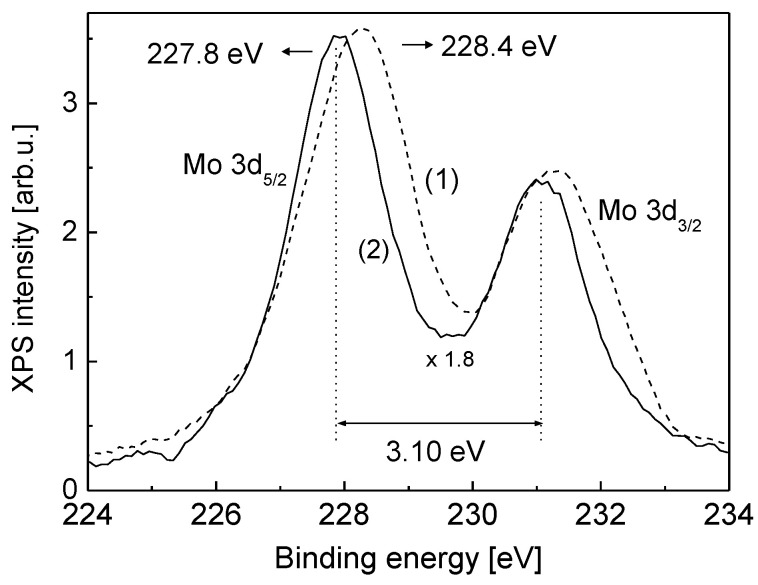
Intensity normalized X-ray photoelectron line of Mo (3d) at room temperature (spectrum 1) and after annealing at a temperature of 1200 K (spectrum 2) of 2 ML boron film on Mo(110).

**Figure 4 nanomaterials-13-00651-f004:**
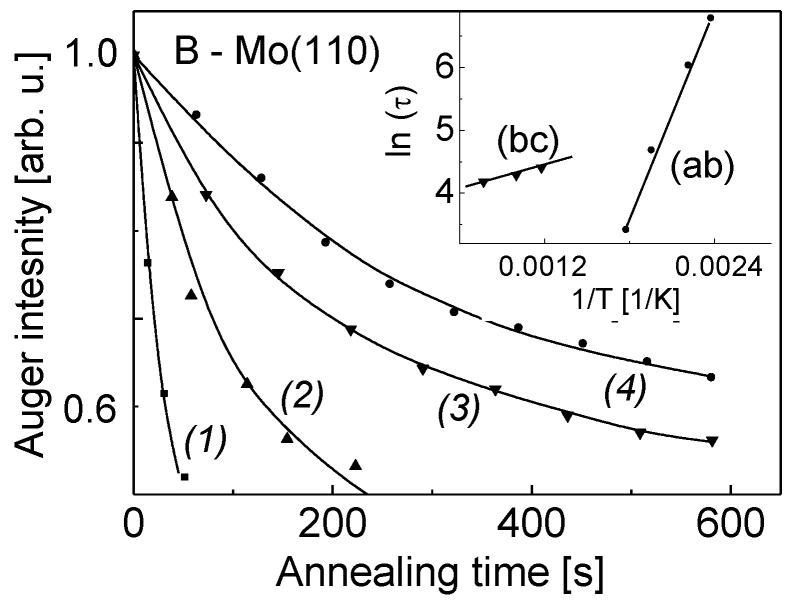
Dependence of the boron KLL Auger intensity on the annealing time for four different similarly prepared B-Mo(110) systems annealed at different temperatures (K): 1–560; 2–510; 3–450; and 4–420 K. Inlay: Dependence of ln(τ) on reciprocal temperature (1/T) for two different annealing temperature intervals corresponding to (ab) and (bc) of Figure 2. The τ is the time required to decrease the initial boron Auger intensity to a certain value (65% of the initial Auger intensity).

**Figure 5 nanomaterials-13-00651-f005:**
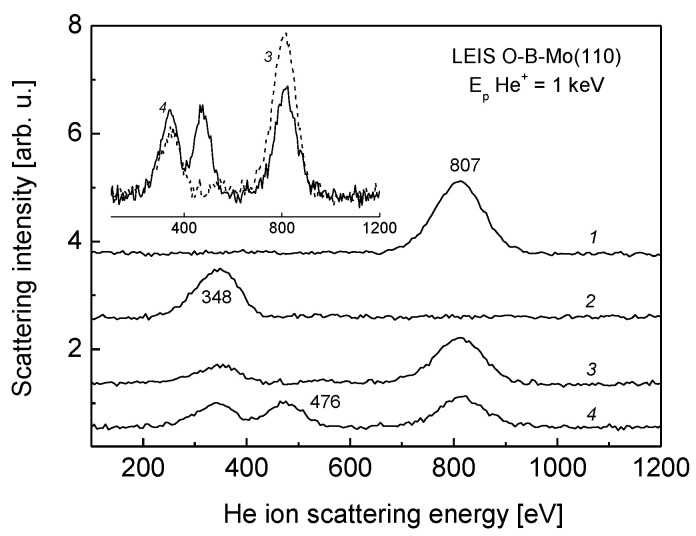
Low-energy ion scattering spectra corresponding to bare Mo(110) before (spectrum 1) and after deposition of 2 ML thick boron film (spectrum 2), molybdenum boride formed after annealing of the boron film at 1200 K (spectrum 3), and the oxide of molybdenum boride (spectrum 4). More visual representation of spectra 3 and 4, demonstrating intensity variation, is shown in an inlay.

**Figure 6 nanomaterials-13-00651-f006:**
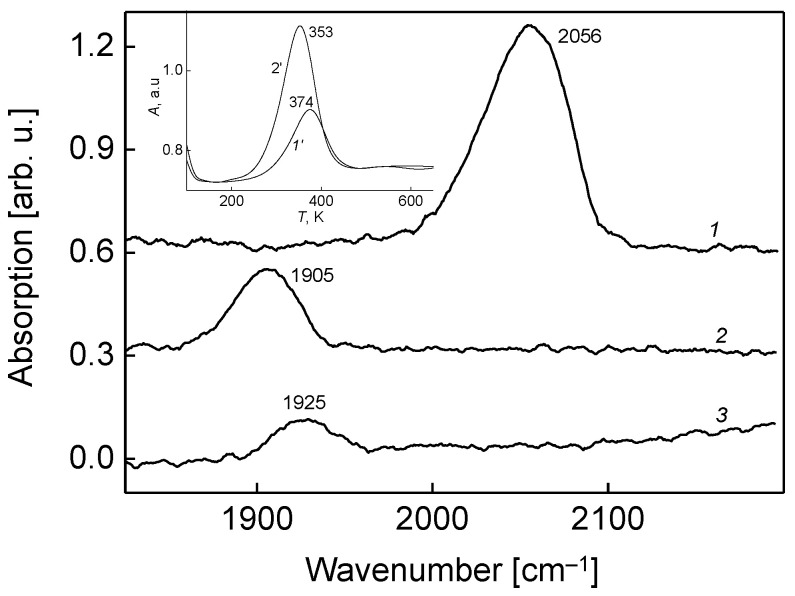
Reflection-absorption infrared spectra of CO molecules adsorbed on the molybdenum boride (spectrum 1), its oxide (spectrum 2) and the surface of the molybdenum boride oxide with pre-adsorbed oxygen at an exposure of 50 L (spectrum 3). In all cases, the substrate temperature is 90 K and the CO exposure is 30 L, which corresponds to saturation coverage. The inlay demonstrates a comparison of temperature-programmed desorption spectra of CO_2_, which is formed via interaction of coadsorbed carbon monoxide and oxygen, from the bare molybdenum boride surface (spectrum 1′) and its oxide (spectrum 2′).

## Data Availability

Data available on request due to privacy restrictions. The data presented in this study are available on request from the corresponding author. The data are not publicly available due to institutional internal regulations.

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
