# Peer review of "Enhancing the Catalytic Activity of Mo(110) Surface via Its Alloying with Submonolayer to Multilayer Boron Films and Oxidation of the Alloy: A Case of (CO + O2) to CO2 Conversion"

_nanomaterials, 2023, doi:10.3390/nano13040651_

Round 1
Reviewer 1 Report
Men et al. report on the boron-deposited Mo(110) surface under O2 and CO+O2 conditions. They provide spectroscopic views of the chemistry over the surfaces using AES, XPS, LEIS, RAIRS and TPD. Overall, the data analysis appears adequate, and the obtained conclusion seems sound. While these results deserve to be published, there are several questions and comments as listed below.
1. I am confusing the data shown in Figure 2. There are too many labels (a-d and 1-3) and arrows in a figure. I could not find the data plot corresponding to (3) and (d). The authors should revise the data presentation in Figure 2.
2. Figure 3: Why does the Mo 3d XP spectrum shift to the lower binding energy side by the boron segregation into the surface? Would the authors explain the origin of the chemical shift?
3. Figure 6: The RAIRS data evidenced that the amount of adsorbed CO on metallic surface (1) was much larger than that on oxidized one (2). On the other hand, the corresponding CO2-TPD spectra showed that the CO2 formation was clearly enhanced rather on oxidized one (2’) than on metallic surface (1’). What is the origin on enhancement of catalytic activity over the oxidized surface? Is only a part of CO on metallic surface active to the reaction?
4. The author should add a scheme (of concluding) figure at the beginning (or end) of the manuscript to clarify the main message of this study.
Author Response
Dear Reviewer,
The comments are gratefully found suitable and helpful.
Below please find point-by-point descriptions of the changes made to the manuscript according to the comments.
Comment: 1. I am confusing the data shown in Figure 2. There are too many labels (a-d and 1-3) and arrows in a figure. I could not find the data plot corresponding to (3) and (d). The authors should revise the data presentation in Figure 2.
Revision: Corresponding revision is made to Figure 2 (line 233).
Comment: 2. Figure 3: Why does the Mo 3d XP spectrum shift to the lower binding energy side by the boron segregation into the surface? Would the authors explain the origin of the chemical shift?
Revision: Corresponding explanation is added to the manuscript (lines 217 --)
Comment: 3. Figure 6: The RAIRS data evidenced that the amount of adsorbed CO on metallic surface (1) was much larger than that on oxidized one (2). On the other hand, the corresponding CO2-TPD spectra showed that the CO2 formation was clearly enhanced rather on oxidized one (2’) than on metallic surface (1’). What is the origin on enhancement of catalytic activity over the oxidized surface? Is only a part of CO on metallic surface active to the reaction?
Revision: Corresponding explanation is added to the manuscript (lines 353 --)
Reviewer 2 Report
The paper “Enhancing the Catalytic Activity of Mo(110) Surface via its Alloying with Submonolayer to Multilayer Boron Films and Oxidation of the Alloy: A Case of (CO+O2) to CO2 Conversion” is interesting, well written and explained. I recommend that the paper be accepted after minor revision. Some recommendations and comments are made below:
Abstract: the authors write”Adsorption and reaction of CO and O2 molecules on the surface of the formed molybdenum boride is dramatically different from the bare Mo(110) surface in terms of higher carbon monoxide oxidation activity.” However, they did not perform themselves the reaction on the Mo(110) surface, only briefly citing a paper on this subject. I think the abstract should contain only experiments done by the authors, not results from the literature.
Introduction
- rows 85-87: “At the same time, the general trend in catalytic activity of materials is that their efficiency increases when shifting from unoxidized materials to the oxides” – citations needed for this claim;
Experimental
- the authors did not present any data that the initial Mo surface showed the (110) plane exposed;
- although TPD is presented as one of the methods used for the characterization of the solids, and also as an inset in Figure 6, there is no description of the method in the Experimental part;
Results and discussion
- Figure 1: instead of numbering the curves and then explaining the numbers in the figure caption, it would be easier to understand the figure (and practically all the figures in the paper) if the curves would have the information on them: 0 ML (or no boron) instead of 1; 0.1 ML instead of curve number 2 and so on;
- Figure 2 and row 212: the c-d part of the curves is not visible, and also curve 3 is not visible. At a first glance the reader believes that curve 3 is the inset of the figure, but then it sees it’s different. Please clarify;
- rows 292-294: “In favor of this assumption is the observed B 1s and Mo 3d5/2 photoelectron lines shift upon oxidation from 190 to 191 eV, and from 228 to 229.5 eV, respectively.” – at what figure are the authors referring?
- rows 364-366: the authors claim “The catalysts prepared in this way and having quite high activity, at least, for CO oxidation, may serve as alternatives to noble metal containing catalysts” – this is a conclusion that cannot be drawn from several tests in a UHV chamber. In normal conditions the catalysts might behave quite differently, and the activity might not be maintained for sufficiently long time so that these solids can be considered as possible alternatives for noble metal catalysts.
Conclusions
- row 385: the authors claim the “appearance of new adsorption/reaction sites at the Mo-B-O interface”. They did not discuss this in the Results and Discussion part (other than a brief mention in row 361) and they should clarify there what these new adsorption/reaction centers are.
There are some English spelling errors in the text:
- rows 36, 139 and 297: “species” instead of “”specie”;
- row 43: “Quite recently they found” instead of “Quite recently they find”;
- row 50: “boronization” (also called boriding) instead of “borodization”;
- row 238: “was plotted” instead of “was plot”;
- row 325: “considerably different” instead of “considerable different”;
- row 340: “is manifested also in the case” instead of “is manifested and in the case”.
Author Response
Dear Reviewer,
The comments are gratefully found suitable and helpful.
Below please find point-by-point descriptions of the changes made to the manuscript according to the comments.
Comment: Abstract: the authors write”Adsorption and reaction of CO and O2 molecules on the surface of the formed molybdenum boride is dramatically different from the bare Mo(110) surface in terms of higher carbon monoxide oxidation activity.” However, they did not perform themselves the reaction on the Mo(110) surface, only briefly citing a paper on this subject. I think the abstract should contain only experiments done by the authors, not results from the literature.
Revision: Corresponding revision is made toe the abstract (lines 31-34).
Comment: - rows 85-87: “At the same time, the general trend in catalytic activity of materials is that their efficiency increases when shifting from unoxidized materials to the oxides” – citations needed for this claim;
Revision: Corresponding citation is added (#10, line 434).
Comment: - the authors did not present any data that the initial Mo surface showed the (110) plane exposed;
Revision: There is corresponding reference to our previous study (#7).
Comment: - although TPD is presented as one of the methods used for the characterization of the solids, and also as an inset in Figure 6, there is no description of the method in the Experimental part;
Revision: Corresponding section is added to the manuscript (lines 114-120)
Comment: - Figure 1: instead of numbering the curves and then explaining the numbers in the figure caption, it would be easier to understand the figure (and practically all the figures in the paper) if the curves would have the information on them: 0 ML (or no boron) instead of 1; 0.1 ML instead of curve number 2 and so on;
Revision: For clarity provided by direct comparison of the spectra, the spectra are plotted against each other within one figure. If put legends by each spectra, the figure would be overloaded. Therefore, only numerical labels with their explanation in the figure caption are used.
Comment: - Figure 2 and row 212: the c-d part of the curves is not visible, and also curve 3 is not visible. At a first glance the reader believes that curve 3 is the inset of the figure, but then it sees it’s different. Please clarify;
Revision: Corresponding clarification is made (line 236,240).
Comment: - rows 292-294: “In favor of this assumption is the observed B 1s and Mo 3d5/2 photoelectron lines shift upon oxidation from 190 to 191 eV, and from 228 to 229.5 eV, respectively.” – at what figure are the authors referring?
Revision: Corresponding referring is added (line 302,303).
Comment: - rows 364-366: the authors claim “The catalysts prepared in this way and having quite high activity, at least, for CO oxidation, may serve as alternatives to noble metal containing catalysts” – this is a conclusion that cannot be drawn from several tests in a UHV chamber. In normal conditions the catalysts might behave quite differently, and the activity might not be maintained for sufficiently long time so that these solids can be considered as possible alternatives for noble metal catalysts.
Revision: Of course, we agree that this is too ambitious a statement, so we toned it down (line 380,381).
Comment: - row 385: the authors claim the “appearance of new adsorption/reaction sites at the Mo-B-O interface”. They did not discuss this in the Results and Discussion part (other than a brief mention in row 361) and they should clarify there what these new adsorption/reaction centers are.
Revision: To properly address this statement further comprehensive studies are to be done. Therefore, this statement is removed from the conclusion section.
Comment: There are some English spelling errors in the text:
- rows 36, 139 and 297: “species” instead of “”specie”;
- row 43: “Quite recently they found” instead of “Quite recently they find”;
- row 50: “boronization” (also called boriding) instead of “borodization”;
- row 238: “was plotted” instead of “was plot”;
- row 325: “considerably different” instead of “considerable different”;
- row 340: “is manifested also in the case” instead of “is manifested and in the case”.
Revision: All have been corrected.
